

# Wind Turbine Gearbox Operation Monitoring Using High-Resolution Distributed Fiber Optic Sensing

Linqing Luo[1], Unai Gutierrez Santiago[2,3], Yuxin Wu[1]

[1]Earth and Environmental Sciences Area, Lawrence Berkeley National Laboratory, 1 Cyclotron Road, MS 74R-316C, Berkeley, CA, 94720, United State
[2]3mE, TU Delft, Mekelweg 2, 2628 CD, Delft, the Netherlands
[3]Siemens Gamesa Renewable Energy, Parque Tecnológico de Bizkaia, 48170 Zamudio, Spain

*Correspondence to*: Linqing Luo (linqingluo@lbl.gov)

**Abstract.** Efficient gearbox monitoring is vital for improving fault detection, enhancing design, and reducing operation and maintenance (O&M) costs, particularly for offshore wind turbines. This paper introduces an innovative approach using high-resolution Distributed Fiber Optic Sensing (DFOS) based on Optical Frequency Domain Reflectometry (OFDR) to measure gearbox strain in real time. By bonding a single optical fiber around the full circumference of a 2.152 m diameter planetary stage in a 3.75 MW wind turbine gearbox, we captured millimeter-scale distributed strain profiles of planetary gears under different input torque levels. Our results show accurate identification of planet gear locations in real time and rotation speed (10.42 revolutions per minute) and a strong linear correlation between applied torque and measured strain across all monitored locations ($R^2 = 0.9997$), with data collected every 2.6 mm. Strain variations of approximately 200 microstrain were observed on individual gear teeth during engagement, providing granular insights into mechanical behavior and load distribution. Additionally, DFOS detected temperature variations during operation, highlighting its capability to concurrently monitor thermal and mechanical anomalies. This study represents the first application of continuous DFOS to a full-scale wind turbine gearbox. The approach offers a scalable and practical solution for early fault detection, improved mechanical performance, and more reliable wind turbine operations, addressing critical challenges in the wind energy sector.

## 1 Introduction

Wind energy is increasingly important worldwide, yet ensuring reliable operation remains challenging, especially offshore. Difficult access and harsh marine conditions significantly raise operations and maintenance (O&M) costs(Stehly and Duffy, 2021). Among the most affected components is the gearbox, which, while essential for efficient power generation, is prone to wear and failure(Peng et al., 2023).

The gearbox alone represents about 13% of a wind turbine's total cost and is a leading contributor to downtime, resulting in expensive repairs and revenue losses due to periods of inactivity(Link et al., 2011). These issues are even more pronounced in offshore wind turbines (OWTs) where limited access and adverse conditions complicate maintenance. Gearbox reliability is thus crucial as most units fail before reaching their 20-year design life(Musial et al., 2007; Sheng, 2013). As of 2014, approximately 1,200 gearbox failures were reported annually(Shell Lubricants, 2017). Beyond direct repair costs,



overheating due to bearing fractures or lubricant degradation can cause severe damage, including turbine fires. In total, O&M expenses can account for nearly 30% of the levelized cost of electricity (LCOE)(European Commission ( Leyre Azcona (CARSA), 2017; Röckmann et al., 2017).

Research has shown that gearbox failures are dynamic (Carroll et al., 2019; Zheng, 2023) and often correlated with rolling element failures(Rommel et al., 2021; Schmidt et al., 2019). The structural complexity of gearbox and their continuous operation in harsh environments make them especially vulnerable(Guo et al., 2018; Huo et al., 2022). Additionally, the failure of each part in a gearbox is not independent, as the interaction of load and strength may affect each other and contributes to gearbox failure (Liu et al., 2022)(Krause et al., 2022).

Currently, early-stage gearbox failures are often detected using CMS (condition monitoring system) signals, including abnormal temperature changes(Feng et al., 2013; Guo and Bai, 2011). Advanced techniques, such as combining spectral kurtosis analysis with hidden Markov models, can identify and classify emerging faults(Li et al., 2020). Additionally, the choice of preventive maintenance intervals affects both reliability and maintenance costs(Igba et al., 2015). While high-speed gears are relatively easy to monitor, planetary stages remain challenging due to their slower rotation and complex load

paths to reach the location where the point-based detection tools are installed. For instance, damage in planetary bearings is difficult to detect using acoustic sensors placed on the ring gear's outer surface.

Existing monitoring techniques rely heavily on point-based sensors spaced sparsely around large gearboxes, which can exceed one meter in diameter. These setups provide limited spatial coverage and struggle to identify small, localized defects(Keller et al., 2017; Rosinski and Smurthwaite, 2010). For instance, strain gauges, a common tool for strain

measurement, offer highly localized data. Collecting multiple strain readings with these sensors complicates wire and data logger management, decreases reliability, and makes field applications impractical. Similarly, incremental encoders (Zhang et al., 2018) can detect relative movement changes but need frequent recalibration and complex analysis, particularly problematic in offshore settings.

Fiber Bragg Grating (FBG) sensors have emerged as an alternative, but they measure only discrete points—usually 10–20

per fiber. For comprehensive coverage of a large gearbox, several fibers and many sensing points are needed, increasing complexity and installation effort. Although Gutierrez Santiago et al. (Gutierrez Santiago et al., 2022) showed that FBG sensors could be applied to gearbox monitoring, the limited sensing points and the need for precise sensor placement remain drawbacks. For example, 4 individual optical fibers with 14 gratings at each fiber must be installed to provide at least 56 points for monitoring purposes. Installing 4 optical fibers and 56 sensing points at predetermined locations were a challenge

due to its large effort on location control of each fiber grating to match sensing locations to the desired surface place.

To overcome these limitations, this study proposes using Distributed Fiber Optic Sensing (DFOS) to measure strain continuously at millimeter-scale resolution around the entire gearbox circumference. By attaching a single standard optical fiber directly onto the gearbox surface, DFOS provides widespread, detailed measurements without the intricate installation design required by point-based sensors. The installation process is straightforward and does not demand specialized training.



With its high-density data and simple setup, DFOS delivers a practical, scalable solution for early fault detection and load
prediction in wind turbine gearboxes and other large industrial equipment.

This paper describes the DFOS for gearbox monitoring methodology and demonstrates the data analysis method to interpret
the gearbox operations. We validate the approach through a laboratory load test on a full-scale wind turbine gearbox using a
back-to-back test bench. The results confirm that DFOS not only offers high-resolution strain data for detecting anomalies
but also reliably estimates gearbox load. This advancement paves the way for more effective and cost-efficient condition
monitoring across the wind industry.

## 2. Gearbox and distributed fiber optic sensor

### 2.1 Wind turbine gearboxes

Wind turbine drivetrains are classified into two main types: direct-drive and geared systems. In direct-drive configurations,
the rotor is directly connected to a low-speed generator. In contrast, geared systems use a gearbox to increase the rotational
speed of the rotor before transferring energy to a medium- or high-speed generator. Both configurations are used by
manufacturers, but recent studies suggest that geared drivetrains may provide a lower LCoE for future large offshore turbines
exceeding 15 MW[27]. A gearbox is a mechanical system that bridges the low-speed rotor shaft and the high-speed generator
shaft, increasing rotational speed to meet generator requirements. The low-speed shaft rotates in sync with the turbine blades,
while the high-speed shaft powers the generator. Gearboxes typically include a combination of planetary and parallel gear
stages, with specific configurations depending on turbine capacity(Gutierrez Santiago et al., 2022). For instance, a single
planetary stage followed by two helical stages (1P2H) is common for turbines up to 2 MW, while two planetary stages
followed by one helical stage (2P1H) are used in turbines ranging from 3 to 8 MW.

This study focuses on epicyclic planetary stages with stationary ring gears. In these stages, the input shaft is connected to the
planet carrier, which houses planet shafts that rotate with the carrier. The planet gears mesh simultaneously with the
stationary ring gear and the central sun gear, increasing velocity between the input carrier and the sun, as shown in Fig. 1.
The gear mesh forces between the planets and the ring gear are proportional to the input torque. Due to the relatively thin
structure of the ring gear, notable deformations could occur on its outer surface, where an optical fiber is installed for
monitoring. By measuring strain on the ring gear, torque and potential abnormalities in gearbox operation can be effectively
tracked.



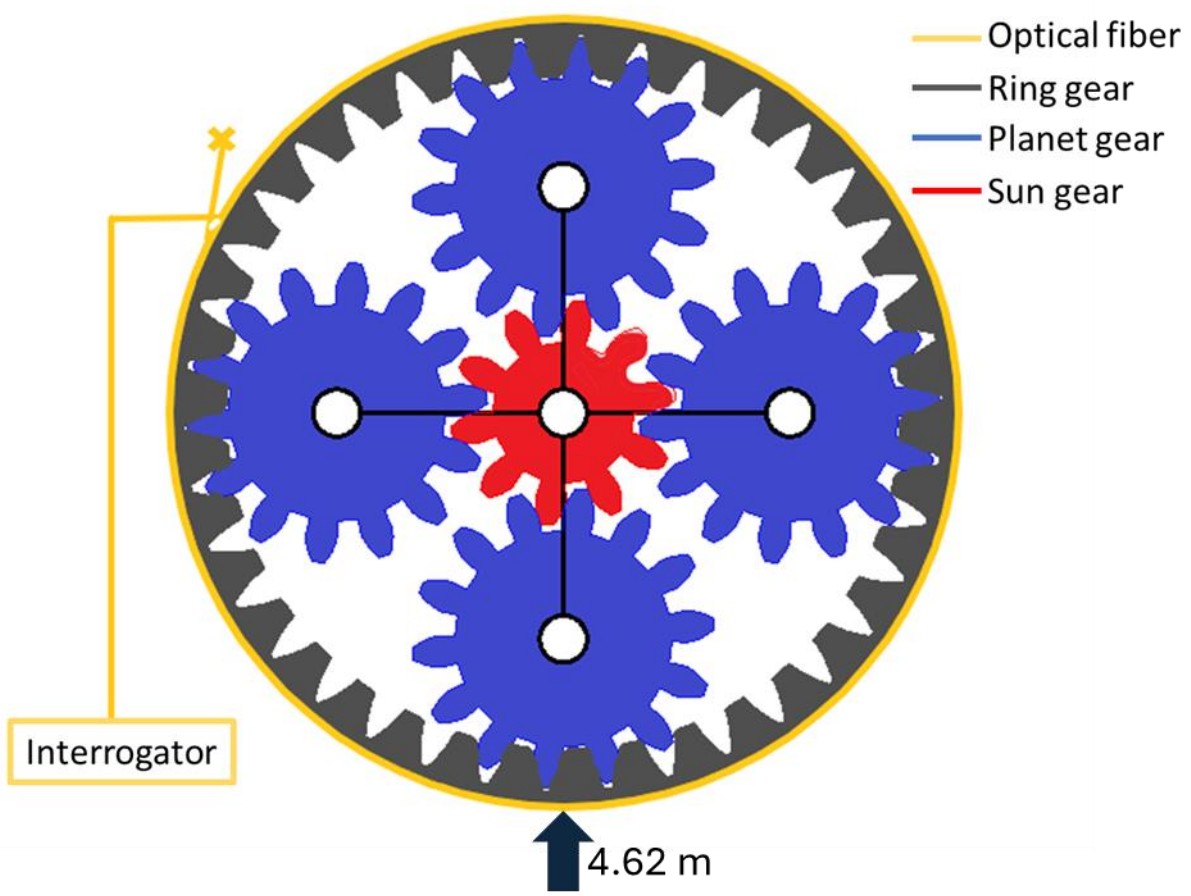

**Figure 1 The planetary gear stage and the optical fiber attached to the outer surface of the gearbox. The black arrow indicates the location at the bottom of the gearbox used for strain profile demonstration. The point at 4.62 m refers to the distance from the starting point of the optical fiber attachment on the gearbox.**

## 2.2 Distributed fiber optic sensing

DFOS systems measure changes in light properties along an optical fiber, enabling continuous sensing at millimeter-scale resolution. This technology relies on naturally occurring light scattering within the fiber, with a small fraction of the light reflected at each point. By analyzing these reflections, strain and temperature changes can be monitored along the fiber's entire length. Two primary methods are used to analyze the scattered light: Optical Time Domain Reflectometry (OTDR) and Optical Frequency Domain Reflectometry (OFDR). OFDR determines the scattering location based on the time-of-flight principle, which relates the tuning frequency and speed of light in the fiber. The scattering location z is calculated using the following equation(Guo et al., 2021; Liang et al., 2021):



$$z = \frac{fc}{4\pi\gamma n} \qquad (1)$$

where z is the location of the event from the position where the pulse is launched into the fiber, c is the light velocity in a vacuum, $f$ is detected tuning frequency, $\gamma$ is the frequency tunning speed, and n is the refractive index of the optic fiber.

In this study, strain and temperature changes were measured with OFDR, providing high spatial and temporal resolution. The frequency shift (Δv) of the scattered light is linearly related to strain and temperature changes as follows:

$$\Delta v = c_\varepsilon \Delta\varepsilon + c_T \Delta T \qquad (2)$$

where $c_\varepsilon$ is the coefficient of strain change, $\Delta\varepsilon$ is the strain change, $c_T$ is coefficient of temperature change, and $\Delta T$ is the temperature change.

This approach enables precise, high-density sensing, allowing for real-time monitoring of strain and temperature variations along the optical fiber. By leveraging these capabilities, DFOS provides detailed insights into gearbox behavior, facilitating accurate fault detection and performance analysis.

## 3. Test setup

### 3.1 Gearbox and Test bench

The tested 3.75 MW gearbox comprises two planetary stages and one helical stage and the first planetary stage has 2.152-meter diameter. Testing was conducted on a back-to-back gearbox test bench, where two wind turbine gearboxes were connected through their low-speed shafts. A motor drove the high-speed shaft of one gearbox, while the high-speed shaft of the second gearbox provided braking load via another motor, simulating real-world wind turbine generator operation.

The first planetary stage includes four planet gears housed in the planet carrier, which serves as the input or low-speed shaft of the gearbox. As the carrier rotates, the planet gears mesh simultaneously with the sun gear and the stationary ring gear. The gear mesh forces are proportional to the input carrier torque and are transmitted through multiple teeth along the contact lines between the planet gears and the ring gear. These forces generate notable strain on the ring gear, dependent on input torque and load sharing among the planet gears.

A 175 µm polyimide-coated single-mode optical fiber was attached to the outer surface of the ring gear in the first planetary stage (Figure 2). The fiber was bonded using Loctite EA 3423 epoxy, providing strong adhesion while remaining removable. This approach allowed the fiber to be detached after testing, as the gearbox was a production model intended for operational deployment. The fiber was connected via an extension cable to the OFDR interrogator, located outside the testing area. The system was used to measure strain and temperature changes during the loading test, with specific setup parameters detailed in Table 1.



**Table 1 The test progress and set up during the loading test.**

| Time | Event | Time | Event |
|---|---|---|---|
| 8:55 am | Data acquisition started | 9:58 am | 17% torque in reversed |
| 9:15 am | Gearbox started (slow) | 10:12 am | 33% torque |
| 9:17 am | Gearbox ramping up speed | 10:16 am | 50% torque |
| 9:18 am | Gearbox nearly stopped | 10:49 am | 66% torque |
| 9:19 am | Gearbox speed up to reach 100% torque to warm up | 11:08 am | 83% torque |
| 9:41 am | Gearbox is stopping | 11:27 am | 100% torque |
| 9:43 am | Gearbox almost complete stop | 12:12 pm | Stop rapidly |
| 9:48 am | Gearbox started in reversed direction with 50% torque | 12:16 pm | Stop recording |
| 9:54 am | Gearbox stopped | | |


### 3.2 Data Acquisition

The test started in the morning following another test using the same gearbox. Before the test, the gearbox ran overnight and stopped to prepare for the test for this study. Data acquisition began before the gearbox started, capturing baseline conditions at slow rotation speeds. The gearbox speed gradually increased, followed by deceleration, to verify the system setup.
Afterward, the gearbox was subjected to a sustained 100% torque load to warm up the system. Subsequently, it decelerated to near-complete stoppage before switching rotation direction to begin the primary ramping load test.

In the ramping test, the gearbox restarted in reverse at 50% torque and then cycled through increasing torque levels: 17%, 33%, 50%, 66%, 83%, and finally 100% of the rated torque. Each percentage represents a torque level normalized to the gearbox's rated capacity. The test concluded with an emergency stop at the maximum torque, signifying the end of the
recording. Detailed test progression and events are summarized in Table 1. A Luna ODiSi 6108 OFDR system was used for data collection, offering high-resolution strain and temperature measurements along the optical fiber. The system's parameters are outlined in Table 2.



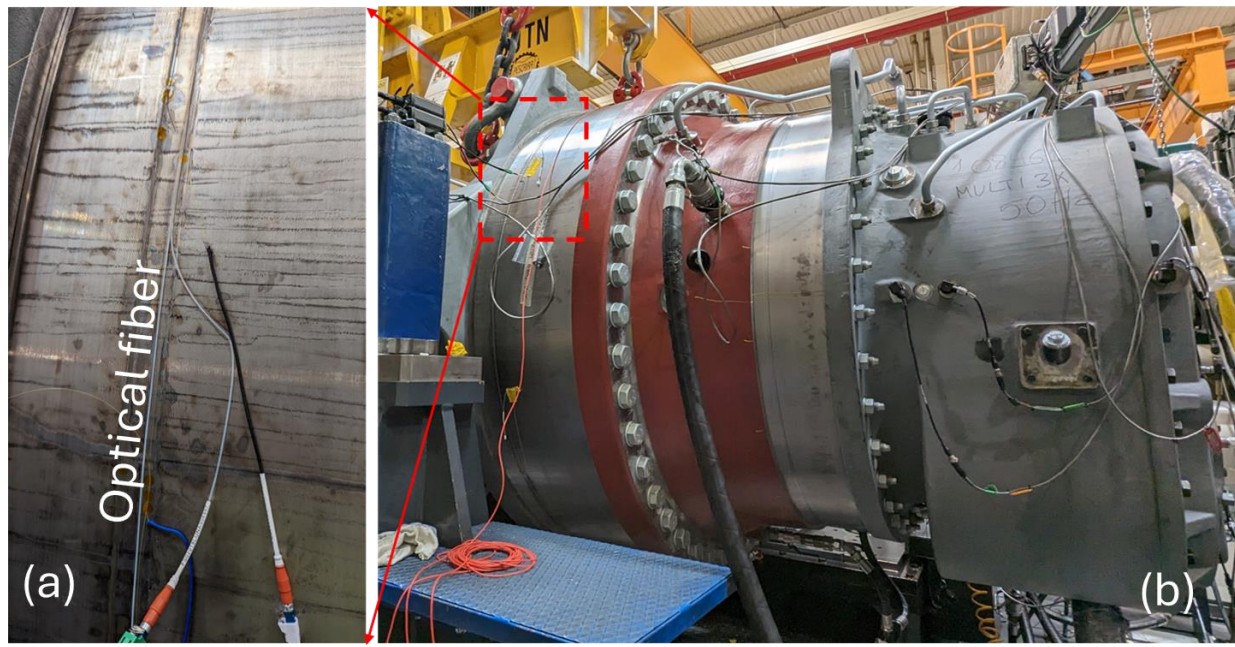

**Figure 2 Photo of the installed optical fibers, one for strain and one for temperature, attached to the outer surface of the gearbox. (a) A close look of the installed optical fibers. (b) The full size of the gearbox.**

**Table 2 The setup of the Luna ODiSi 6108 for data collection**

| Parameter | OFDR value |
| --- | --- |
| Measurement frequency | 12.5 Hz |
| Spatial Resolution | 2.6 mm |
| Optical fiber length | 10.2948m |
| $c_\varepsilon$ strain coefficient | -0.638 micro-strain/GHz |
| $c_T$ Temperature coefficient | -6.67 ℃/GHz |





## 4. Results and Discussion

### 4.1 Raw data overview

The raw strain data collected during the test are shown in Figure 3a, where red indicates tensile strain and blue represents compression. The diagonal banding pattern corresponds to gear rotation, with rotation direction changes visible in the shifting patterns between 9:19 to 9:41 (clockwise) and 9:48 to 12:12 (counterclockwise). Figure 3b provides a detailed view of a 30-second interval outlined by white dashed lines in Fig.3a, revealing the distinct locations of the four planet gears as

they rotate along the circumference of the ring gear.

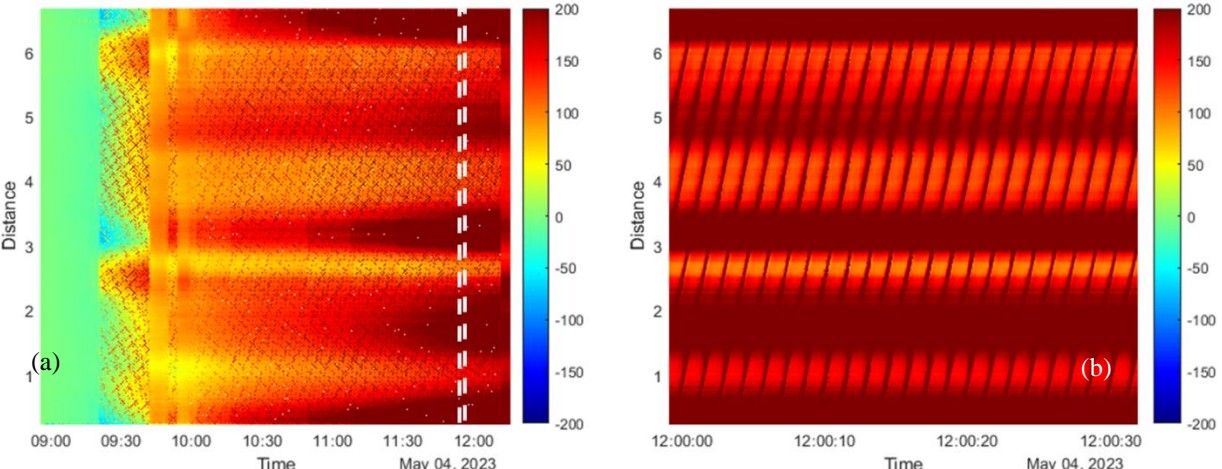

**Figure 3 Waterfall representation of the strain profile over time for the circumference of the ring rear. The vertical axis denotes distance in meters, while the color gradient indicates strain in micro-strain. (a) Strain profile throughout the entire test period. (b) A magnified view of the strain profile corresponding to the region outlined by the white dashed square in the left panel.**

As the test progressed, a shift toward darker red regions indicated increasing strain, which was attributed to rising temperatures from frictional heat generation between the gears. The strain distribution and rotational behavior of the gears could be clearly tracked throughout the test.

### 4.2 Temperature Effects

Temperature changes were evident in the strain data, as shown in Figure 4 , which depicts the response at a single location

(bottom of the gearbox in Figure 1 or 4.62 m in Figure 3). Before 9:15 am, the strain exhibited a downward drift of approximately 30 microstrain, corresponding to a cooling effect of about 3°C from the gearbox's overnight test. After 9:15 am, strain rose by approximately 150 microstrain, equivalent to a 15°C temperature increase, caused by heat generated during operation. The maximum drift rate occurred between 9:19 and 9:41 am, coinciding with the transition from idle to maximum speed. Temperature effects were removed using a 0.01 Hz high-pass filter to isolate dynamic strain variations




from slower thermal processes. The filtered data, shown in Figure 5 and Figure 6, highlights the strain behavior under varying loads.

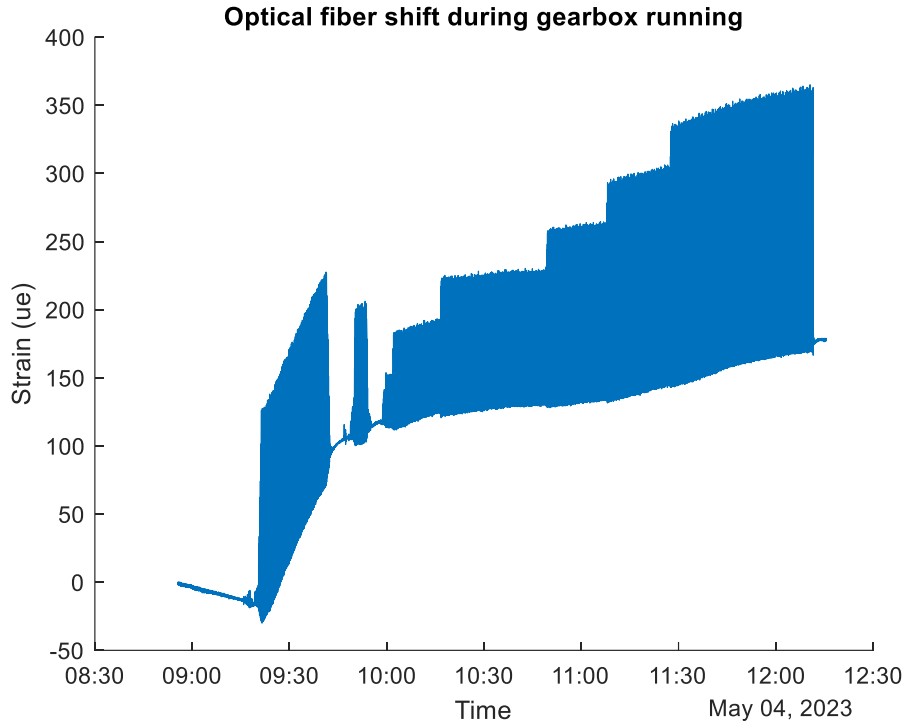

**Figure 4 The strain response at a random selected location (4.62m) during the loading test.**

### 4.3 Gear Engagement and Torque Correlation


The strain profile during the maximum load test (Figure 5) reveals repeated tensile spikes forming a diagonal pattern, indicating the rotation of the four planet gears. At each timestamp, four red bands (tensile strain) correspond to the locations where planet gears contact the ring gear. These are separated by mild blue bands (compressive strain) where there is no direct gear contact.

After 9:48 am, both tensile and compressive strain intensities increased as the torque rose from 17% to 100%. Each tooth on the ring gear exhibited cyclic strain patterns, reflecting the load distribution during gear engagement. The strain width across approximately five teeth suggests contributions from the helix angle of the gear teeth and partial strain transfer before and after full contact. At the bottom of the gearbox (4.62 m), strain values increased from 33 microstrain at 17% torque to 161 microstrain at 100% torque (Figure 6). These measurements demonstrate how DFOS captures detailed strain variations, even

at the level of individual gear teeth.

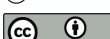



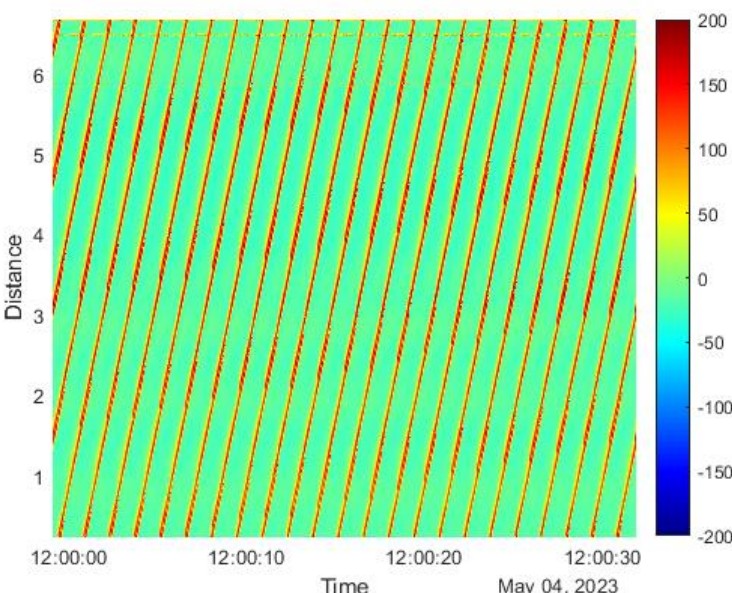

**Figure 5 The waterfall of the strain response during the load test after 0.01 Hz high pass filter in period of 30 seconds when the load is at 100% torque, time domain shows strain profile collected with 12.5 Hz.**

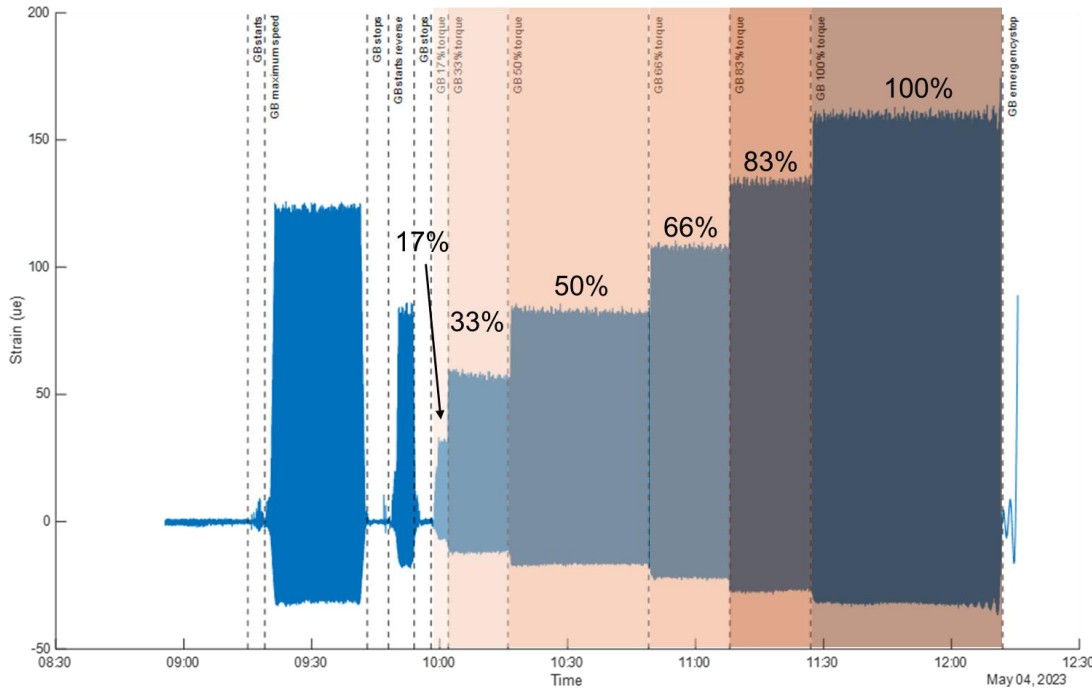

**Figure 6 The strain response after 0.01 Hz high pass at location 4.62m (one tooth's strain response). The dash line shows the timing of each loading event listed in table 2. The color background shows the percentage of the rated torque input from 17% to 100%.**





Examples of strain profiles at various torque levels are shown in Figure 7a, plotted on a polar axis to illustrate the gearbox's real-time response. The four prominent strain spikes correspond to the locations of the planet gears, with these spikes

rotating as the gears move. This polar axis plot can be generated in real time for each strain profile measurement, clearly displaying the positions of the planet gears and the strain values they produce. A supplemental video illustrating the dynamic rotation of the planet gears and associated strain spikes is in the supplemental documents (Luo and Wu, 2024).

As the gears rotate, six strain profiles corresponding to different torque inputs are presented, with the time frames selected to align with consistent gear-tooth engagement. At higher torque levels, tensile strain increases at points where the planet gears

contact the ring gear, while compressive strain intensifies in areas between gear contacts. This behavior is more clearly depicted in Figure 7b, which shows strain values recorded at a specific tooth location (4.62 m). The data includes strain measurements as the gear meshes with and disengages from the tooth, highlighting the cyclic strain variations under different torque conditions.

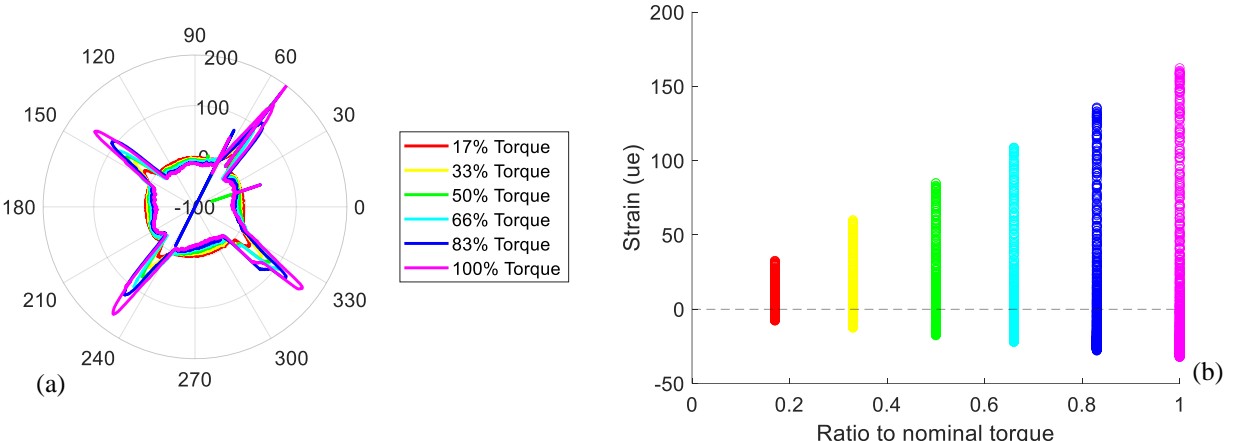

**Figure 7 (a) The strain profile in polar axis when different loads are applied to the gearbox and (b) the scatter plot show the clusters of all the collected strain at one location on the ring gears during the load test with different load input.**

The strain variation at five specific teeth is illustrated in Figure 8a. Each distinct spike corresponds to the arrival and departure of a planet gear during its rotation cycle. The time interval between consecutive spikes is measured at 1.44 seconds, indicating a full rotation duration of 5.76 seconds for the first stage. This translates to a rotational speed of 10.42

revolutions per minute (rpm). Additionally, a periodic signal comprising six spikes over 8.64 seconds is observed at all measured locations. This pattern likely reflects periodic fluctuations in the input torque from the back-to-back gearbox system used during testing.

**4.4 Linear Response with Torque**

To analyze the relationship between strain and torque, the peak-to-peak strain was calculated for each measurement location.

The peak-to-peak strain, defined as the difference between maximum tensile and maximum compressive strain, captures the



full range of strain variation at a given tooth. This metric was computed for 49 evenly spaced locations along the ring gear circumference to assess the strain response under varying torque levels. The results revealed a linear relationship between torque and peak-to-peak strain, as illustrated in Figure 8b. The average linear relationship is shown in Eq. 3. This linear correlation was exceptionally strong, with a high coefficient of determination ($R^2$=0.9997).

It's important to note that the gearbox used for this test was newly manufactured and this testing was conducted in conjunction with the factory's quality control process. However, as the gearbox ages, deviations from this linearity may appear due to factors such as material fatigue, ratcheting, inadequate lubrication, or changes in gear material thickness. Monitoring these deviations over time could provide critical insights for fault detection and predictive maintenance, enabling proactive interventions to prevent failures.

$$Strain = 184.57 \times Load\ percentage + c \qquad (3)$$

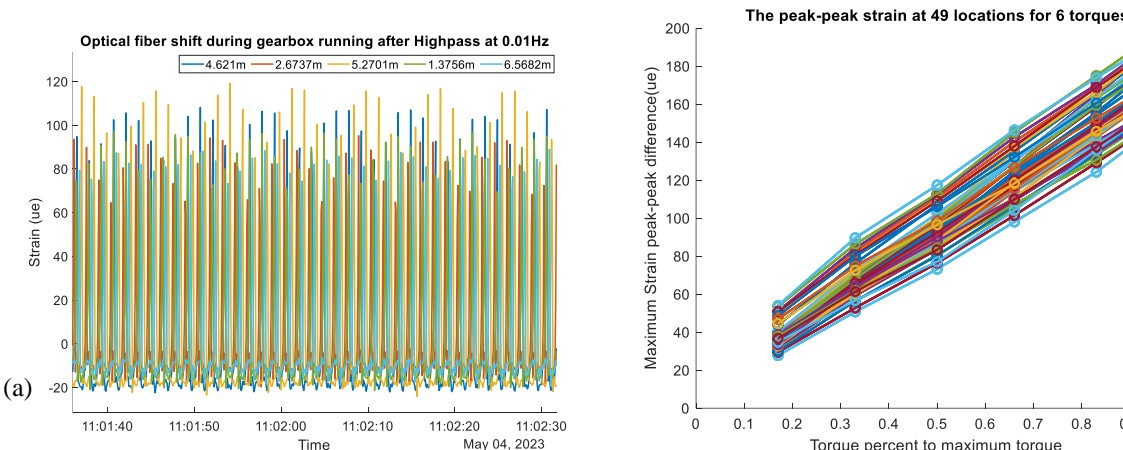

**Figure 8 (a) An example of collected strain at 5 location with time and (b) the peak-peak strain at 49 locations on the ring gear for 6 torque load inputs.**

## 5. Discussion

The DFOS system offers high spatial resolution, with a 2.6 mm resolution that enables precise strain measurement along the entire circumference of the gearbox. This resolution is sufficient to accurately capture the strain response at each gear tooth. This study validates that strain variations serve as reliable indicators of torque load in wind turbine gearboxes, aligning with findings from previous research using Fiber Bragg Grating (FBG) sensors (Gutierrez Santiago et al., 2022). A strong linear relationship between torque and strain was observed across the distributed sensing range, showcasing the potential of DFOS

for real-time condition monitoring. By enabling full circumferential monitoring of the gearbox under dynamic load conditions, DFOS can help localize the planet gears, identify minor anomalies, facilitating proactive maintenance and reducing the risk of significant damage. This capability is particularly valuable for offshore wind farms, where minimizing O&M costs is a priority.





The ability of DFOS to produce high-resolution strain data in real time introduces several benefits for wind turbine gearboxes:

- **Comprehensive Coverage**

A single optical fiber can monitor the entire circumference of the ring gear, providing unprecedented spatial coverage compared to traditional point-based sensors. Unlike strain gauges or FBG sensors, which measure strain at discrete points, DFOS enables continuous sensing along the fiber, capturing data at 2.6 mm, or better, spatial resolution. This level of detail ensures the detection of localized anomalies such as micro-cracks, uneven gear wear, or contact misalignment.

Furthermore, DFOS allows for simultaneous monitoring of all gear teeth as they engage with planet and ring gears, offering a complete picture of the mechanical behavior under varying load conditions. This comprehensive view supports not only fault detection but also theoretical modeling of physical behaviors and insights for future gearbox design improvements.

- **Early Fault Detection**

The high-resolution and high-sensitivity strain data provided by DFOS makes it possible to detect subtle deviations from baseline strain patterns that may indicate the onset of mechanical issues. For example, the system can identify changes in strain response associated with uneven load distribution, gear misalignment, material fatigue, or insufficient lubrication. These deviations often preceded more serious failures, allowing maintenance teams to intervene early and prevent escalation. By monitoring the strain profile over time, DFOS can detect evolving issues, such as gradual wear or the development of cracks, which may not be immediately apparent through conventional monitoring techniques. Moreover, the linear relationship between torque and strain observed in this study provides a baseline for detecting irregularities. Deviations from this linearity at specific locations can serve as early warning signs of anomalies such as localized stress concentrations or weakening material.

Early fault detection is particularly valuable in offshore wind turbines, where maintenance operations are costly and challenging due to limited accessibility and harsh environmental conditions. The ability to identify potential issues during regular operation, rather than relying on scheduled inspections or reactive maintenance, significantly reduces downtime and associated costs. Furthermore, this capability supports a transition from time-based to condition-based maintenance strategies, optimizing resource allocation and extending the overall lifespan of the gearbox.

- **Practical Installation**

Installing a single standard optical fiber is less labor-intensive and reduces the risk of sensor failure compared to deploying multiple discrete sensors. DFOS can also concurrently capture temperature data, providing an additional indicator of potential issues such as overheating or lubrication breakdown.

- **Scalability**

While this study measured strain using a single optical fiber installed on the planetary gear, DFOS systems can scale to monitor multiple fibers simultaneously. This enables comprehensive monitoring of the entire gearbox and potentially the nacelle, extending the system's applicability beyond individual components.



These features support comprehensive monitoring of the gearbox, enabling proactive maintenance and improving operational efficiency. This is particularly important for offshore wind farms, where reducing downtime and O&M costs is critical.

Despite its advantages, some minor data artifacts were observed, such as sharp spikes at fixed locations (e.g., 18.7° and 63° in Figure 7a). These irregularities are likely caused by micro-bending of the optical fiber during installation or the application of epoxy. Refining installation techniques and applying advanced post-processing methods, such as outlier fitting, could mitigate these issues in future applications.

Although temperature effects were filtered out in this analysis to focus on strain behavior, the system's capability to measure temperature concurrently should not be overlooked. As shown in Fig. 4, DFOS captured a potential 3-degree cooling when the gearbox stopped rotating and 15-degree heating during gearbox operation at the bottom of the gearbox. If localized overheating occurs, such as from internal damage or inadequate lubrication, the system can detect and pinpoint these anomalies, preventing potential thermal runaway and costly damage. This adds an additional dimension to gearbox condition monitoring.

For this study, a single optical fiber was installed on the first planetary stage. Practical applications could extend this setup to include longer fibers or multiple fibers distributed across various gearbox stages. However, monitoring high-speed rotational components is constrained by the current system's sampling rate of 12.5 Hz. Enhancing OFDR technology to support faster measurement speeds, such as 120 Hz or more, will be critical for monitoring high-frequency operational stages and achieving comprehensive condition monitoring for modern gearboxes.

## 6. Conclusion

This study introduces a novel approach to measuring fully distributed strain profiles along a planetary gearbox using DFOS. Leveraging OFDR technology, DFOS achieves millimeter-scale spatial resolution and a strain measurement accuracy of approximately 1 microstrain at a sampling rate of 12.5 Hz, with approximately 2,500 data points collected simultaneously around the gearbox. The approach was validated through strain measurements conducted on the first planetary stage during a load test. By continuously bonding a standard optical fiber to the outer surface of the ring gear, DFOS enabled the acquisition of a comprehensive strain profile as the planet gears rotated.

The load test results demonstrated that strain increased linearly with applied torque across all monitored locations, highlighting the accuracy and reliability of DFOS. Specifically, the system successfully detected the following key parameters:

- The planetary gears operated at a rotational speed of 10.42 revolutions per minute.
- A strong linear relationship was observed between the applied torque and measured strain, providing a baseline for condition monitoring in a newly manufactured gearbox.
- DFOS effectively monitored the ring gear's rotational direction, strain response, and potential temperature variations in real time.

These findings confirm that DFOS can be used to accurately quantify applied torque and detect irregular strain patterns,
enabling early identification of potential faults before they escalate into catastrophic failures. The high spatial resolution of
DFOS also allows for detailed tracking of strain behavior at each gear tooth during meshing. This capability can contribute
to optimizing tooth design for improved mechanical performance and refining gearbox control systems. Additionally, the
straightforward installation process reduces the need for specialized training and accelerates deployment, significantly
lowering operation and O&M costs.

While this study focused on strain behavior, DFOS also offers valuable insights into temperature variations, similar as
condition monitoring systems but with higher data density. This capability can assist in fault prediction by identifying heat
anomalies, such as those caused by inadequate lubrication or internal damage, providing a comprehensive view of the
gearbox's operational health.

Despite its advantages, the current DFOS implementation is limited by its measurement speed, which may be insufficient for
monitoring later-stage components operating at higher rotational speeds. Enhancing DFOS technology to achieve a sampling
rate of at least 120 Hz would enable more comprehensive monitoring, allowing a single optical fiber to cover the entire
gearbox under high-speed operational conditions.

In summary, this research highlights the novelty and potential of DFOS as a transformative tool for real-time, high-
resolution monitoring of wind turbine gearboxes. By enabling accurate strain and temperature measurements, DFOS not only
enhances fault detection but also supports more efficient gearbox design, operation, and maintenance. Future advancements
in system speed and scalability will further expand their applications, contributing to the development of reliable and cost-
effective wind energy solutions.

**Acknowledge**

This work was supported by the EPIC (Electric Program Investment Charge) program of California Energy Commission
(CEC). We would like to thank Siemens Gamesa Gearbox Facility and staff in Lerma, Spain for access to the gearbox and
assistance during the experiments.

**Author contribution**

Linqing Luo, Unai Gutierrez Santiago, and Yuxin Wu designed the experiments and Unai and Yuxin carried them out.
Linqing performed data processing, and all authors contributed on data analysis. Linqing prepared the manuscript with
contributions from all co-authors.




## Competing interests

The authors declare that they have no conflict of interest.

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
