# Peer review of "Wind Turbine Gearbox Operation Monitoring Using High-Resolution Distributed Fiber Optic Sensing"

_Wind Energy Science, 2025_

## Author Response (AR1)

**Original Manuscript ID:** wes-2025-1

**Original Article Title:** "Wind Turbine Gearbox Operation Monitoring Using High-Resolution Distributed Fiber Optic Sensing"

**To:** Wind Energy Science

**Re:** Response to reviewers

Dear Editor,

Thank you for allowing the resubmission of our manuscript, with an opportunity to address the reviewers' comments.

We appreciate the detailed and constructive comments and questions from the reviewers. We hope the revised manuscript addresses all the comments.

Best regards,

Linqing Luo on behalf of the authors

**Reviewer#1, Comment # 1: Lines 12-14. Without specifying that the fiber was bonded to the outer surface of the ring gear, some readers may be confused. I recommend adding "…of the outer surface of a 2.152 m diameter ring gear of the first planetary stage in a…" and "…distributed strain profiles of the ring gear from planet gear passage under different torque levels."**

**Author response:** The authors appreciate the reviewer's comment and modify the manuscript accordingly.
* * *
**Reviewer#1, Comment# 2: Line 16: I recommend slightly modifying the statement "…200 microstrain…on individual gear teeth during engagement". I believe this refers to Figures 4 and 6; however, it's not clear from the density of the data if this is at the frequency of the planet-ring gear mesh, or of the planet passage. I think it's likely the latter, but I'm not sure. As worded with "on gear teeth", it may also confuse the reader. I recommend this be rephrased to simply "…200 microstrain were observed on the ring gear exterior from each (planet passage or planet-ring gear mesh) event".**

**Author response:** The authors appreciate the reviewer's comment and modify the manuscript accordingly.
* * *
**Reviewer#1, Comment # 3: Line 18: The sentence about detecting temperature variations should be revised, as I believe the claim here may be exaggerated at this point. Basically, although vague temperature "variations" may have been measured, that doesn't necessarily mean the system is "capable" of monitoring "thermal and mechanical anomalies". Really, this is just a hypothesis or potential application of the system, rather than a demonstrated capability as far as I can tell from the article.**

**Author response:** The authors appreciate the review's comment and agree that temperature detection is a hypothesis in this article. We have changed the manuscript from

*"Additionally, DFOS detected temperature variations during operation, highlighting its capability to concurrently monitor thermal and mechanical anomalies."*

To

*"Additionally, DFOS also showed its potential to detect temperature variations during operation, indicating its potential to concurrently monitor thermal and mechanical anomalies."*
* * *
**Reviewer#1, Comment # 4: Line 21: Can the authors elaborate on what they mean by "improved mechanical performance"?**

**Author response:** Thank you for this comment. We originally used the term "improved mechanical performance" to indicate that our sensing method offers valuable feedback on the planetary gear design. In our study, we demonstrated this concept by installing a single loop of optical fiber. In future or practical applications, the optical fiber could be arranged in multiple loops to monitor various locations on the ring gear. This approach would generate a richer dataset for design validation. When combined with modeling, these data can help enhance the ring gear's structural integrity and operational efficiency, leading to a more durable and optimized design.

This sentence has been modified to:

*"The approach offers a scalable and practical solution for early fault detection, enhancing the ring gear's structural integrity and operational efficiency, addressing critical challenges in the wind energy sector."*
* * *
**Reviewer#1, Comment # 5: Lines 25-26: Following the previous sentences and as written, this sentence implies the gearbox is "most affected" for offshore turbines. I'm not aware that such reliability data exists in the public domain – certainly Peng (2023) does not include this, at least not that I can find in it. I recommend the authors revise this sentence. To be honest, Haus et al (2024) at https://app.box.com/s/ktjzjdxn77omu1cjoy9znymbrynlsw0d is a more valuable reference as it compares replacement statistics for gearboxes, generators, and pitch bearings. Technically, the gearbox is not "essential" for power generation either – obviously those that make direct-drive turbines would disagree for the very reasons stated here.**

**Author response:** Thank you for highlighting these issues. We agree that the original wording may overstate the role of the gearbox and imply that it is "most affected" in offshore turbines based solely on Peng (2023). In addition, we acknowledge that the gearbox is not indispensable for all turbine designs, as direct-drive systems do not utilize gearboxes. In light of your suggestion, we have revised the text and incorporated Haus et al. (2024) to better support our discussion on component replacement statistics. The revised passage now reads:

*"Wind energy is increasingly important worldwide, yet ensuring reliable operation remains challenging, especially offshore. Difficult access and harsh marine conditions significantly raise operations and maintenance (O&M) costs (Stehly and Duffy, 2021). Among several critical components, gearboxes have been observed to experience notable wear and failure (Haus et al., 2024; Peng et al., 2023)."*
* * *
**Reviewer#1, Comment # 6: Line 27: A reference for the 13% cost would be helpful. I recommend Stehly and Duffy (2021) https://www.nrel.gov/docs/fy22osti/81209.pdf. Having said that, by this reference the entire drivetrain assembly (i.e. main bearing, main shaft, gearbox, coupling, and I believe also the generator) is 13% of the capital cost of a 2.8 MW reference turbine, not just the gearbox. I don't have a good reference for gearbox cost alone, but it is likely around 10%.**

**Author response:** The authors appreciate the comment. The suggested modification has been added to the manuscript.
* * *
**Reviewer#1, Comment # 7: Line 30: Rather than the very old Musial (2007) and Sheng (2013), the earlier suggested reference of Haus et al (2024) serves the same purpose and is much better.**

**Author response:** The authors appreciate the comment. The suggested modification has been added to the manuscript.
* * *
**Reviewer#1, Comment # 8: Line 31: I was not able to find the reference for Shell Lubricants: IMPROVING WIND TURBINE GEARBOX RELIABILITY, 1–12 pp., 2017. I recommend this reference simply be deleted as the other references are electronically available. If desired, such a replacement statistic could be derived from Haus et al (2024), as a typical replacement rate is around 2% per year for on the order of 100,000 wind turbines in the US (not worldwide) = 2,000 gearbox replacements per year in the US alone.**

**Author response:** Thanks for the comment. The old reference has been deleted and new reference has been added to the manuscript.

**Reviewer#1, Comment # 9: Line 46: I'm not aware that it's common that fielded CMS systems use acoustic sensors. Wouldn't vibration sensors be better here?**

**Author response:** We appreciate the reviewer for the suggestion. The manuscript has been modified as:

*"For instance, damage in planetary bearings is difficult to detect using acoustic or vibration sensors placed on the ring gear's outer surface."*
* * *
**Reviewer#1, Comment # 10: Lines 58-60: My understanding of this work described here is that the example gearbox was over-sensored for test purposes. That is, I don't believe that the gearbox "must" have been instrumented with this number of sensors. It would be more accurate to simply say "…at each fiber were installed…".**

**Author response:** We appreciate the reviewer for the suggestion. Yes, this was a not "must" but just a suggestion. The manuscript has been modified accordingly.
* * *
**Reviewer#1, Comment # 11: Lines 62-64: Is the sentence here intended to contrast DFOS to normal FOS, or DFOS to only strain gauges and/or incremental encoders when it uses the phrase "point-based sensors"? That is, normal FOS requires a similar installation to DFOS.**

**Author response:** We appreciate the reviewer's suggestion. This comment pertains to both strain gauges/incremental encoders and conventional FOS systems. Conventional FOS have predetermined locations on the optical fiber for sensing strain or temperature. Consequently, the exact positions of these sensing points must be known during manufacturing so that the fiber can be installed with these sensing locations aligned correctly. This process requires both careful designs to determine the optimal sensing locations and skilled installation.

In contrast, the distributed sensing capability of DFOS automatically identifies sensing points along the fiber. As a result, only the loop installation location needs to be specified, making the installation process significantly faster and simpler than that for strain gauges or conventional FOS.
* * *
**Reviewer#1, Comment # 12: Line 69 and 115: Again, it is only a hypothesis at this point that such a system can "detect anomalies" or that it facilitates "accurate fault detection". If I misunderstand and it is well-established that strain sensing can detect typical gear and bearing failures in a gearbox, please include a citation. If not, then instead of "…for detecting anomalies…" I recommend "…with the potential of detecting anomalies…". I see that this is done later in Line 89, where the phrase "potential abnormalities" is used.**

**Author response:** The authors appreciate this comment. The manuscript has been changed as suggested.

*"The results confirm that DFOS not only offers high-resolution strain data with the potential of detecting anomalies but also reliably estimates gearbox load."*

and

*"By leveraging these capabilities, DFOS provides detailed insights into gearbox behavior, facilitating the potential for accurate fault detection and performance analysis."*
* * *
**Reviewer#1, Comment # 13: Line 125: I recommend reference of previous works by Gutierrez-Santiago et al (2022) (already cited) and 2024 (https://iopscience.iop.org/article/10.1088/1742-6596/2767/4/042022) and 2025 (CWD 2025 paper, due out any day in Forschung im Ingenieurwesen) be added to this sentence.**

**Author response:** We appreciate the comment by the reviewer. The suggested citations have been added to the manuscript.
* * *
**Reviewer#1, Comment # 14: Line 165: I'm not sure I understand attributing increasing strain to only "rising temperatures from frictional heat generation between the gears". I would expect the gearbox heated up during the test period, but the torque loads also increased from 33% to 100% in the period from 10A to 12P. Per equation 2, aren't both important? Which effect might be most dominant in the strain? Maybe I simply misunderstand this sentence. Or is the evidence for this statement described in the next section? It appears in figure 4 that temperature effects are responsible for an increase in strain of about 150 ue, while torque effects are responsible for up to 200 ue.**

**Author response:** We appreciate reviewer's comment. Sorry for the confusion. The strain mentioned here was derived directly from the measured frequency shift without temperature compensation, therefore, the strain is actually a combination of temperature and strain. To eliminate the confusion, we modified the manuscript.

The manuscript was modified as:

*"As the test progressed, a shift toward darker red regions indicated an increase in measured strain. However, since this strain was derived directly from the frequency shift, it still reflected a combination of both temperature and mechanical effects. The darker red horizontal strips were attributed to rising temperatures caused by frictional heat generated between the gears, while the diagonal strips were linked to the mechanical strain from the meshing of the rotational gears. Overall, the strain distribution and rotational behavior of the gears were clearly tracked throughout the test."*
* * *
**Reviewer#1, Comment # 15: Line 227: Can the authors explain what they mean by "ratcheting"? I am not familiar with this term. In a similar fashion, I'm not sure I understand "changes in gear material thickness" – this I interpret to mean tooth wear such as micropitting.**

**Author response:** We thank the reviewer for catching this. Ratcheting refers to the progressive, incremental deformation that occurs in a material when it is subjected to cyclic loading. Even if each individual cycle applies stress below the material's yield strength, the repeated loading can lead to an accumulation of plastic strain over time. In gearbox, the tooth mesh and thermal cycle also have cyclic small load, therefore, deformation can also happen. Yes, the changes in gear material thickness meant tooth wear such as micro-pitting. The manuscript has been modified according to the suggestion.
* * *
**Reviewer#1, Comment # 16: Line 238: I recommend a citation to Gutierrez Santiago et al. (2025) be added here.**

**Author response:** The authors appreciate this comment, and the citation has been added to the manuscript.
* * *
**Reviewer#1, Comment # 17: Line 250: As before, I recommend a citation be added related to strain sensing of such faults if it exists, or the statement "ensures the detection" be softened to "…detail can potentially detect localized anomalies such as…". Here micro-cracks are specifically referred to, whereas later cracks are referred to. I can foresee that a large through crack in the ring gear could be detected, but "micro-cracks" described here seem a stretch.**

**Author response:** Thanks for this comment. The manuscript has been modified as suggested. The optical fiber can be used to detect cracks around 0.1mm. If the optical fiber can be applied to wrap many loops around the ring gear, it has potential to detect small cracks. However, the author agrees that this experiment did not prove this, and the micro-crack has been removed from the manuscript.

**Reviewer#1, Comment # 18: Line 256 and 259: Again I recommend a citation, or the statement be softened from "can identify" to "may be able to identify" and "can detect" to "may be able to detect". Certainly I can foresee a full crack in the ring gear could be detected.**

**Author response:** We appreciate the comment by the reviewer. The suggestion modification has been added to the manuscript.
* * *
**Reviewer#1, Comment # 19: Lines 279-282: These sentences describe something that I don't believe has been mentioned yet, and appear to be mis-placed in the "Scalability" section. This seems much more natural to be a caveat in the "Practical Installation" section, but should also be discussed earlier with Figure 7a.**

**Author response:** Thanks! This was not intended to be in the "Scalability" section. This was intended to be the discussion of the limitations of the current DFOS system.

A sentence was added to the description of Figure 7A:

" *The sharp spikes observed at fixed locations (e.g., 18.7° and 63°) exhibit reading errors and constant values, likely due to poor bonding of the optical fiber to the ring gear.*"
* * *
**Reviewer#1, Comment # 20: Lines 283-288: Similarly, these sentences also seem to be mis-placed in the "Scalability" section. This seems much more natural to be in the "Early Fault Detection" section.**

**Author response:** We appreciate this comment. A sentence highlights the start of discussion of limitations have been added to the manuscript:

"Despite its advantages mentioned above, the current DFOS system also have limitations."
* * *
**Reviewer#1, Comment # 21: Line 305: I'm not sure I understand the connection between applied torque, measured strain, and providing a baseline for condition monitoring. I believe "providing a means to measure gearbox usage" is the sentiment of the latter part of the sentence.**

**Author response:** We appreciate this comment from the reviewer. The sentence has been changed to:

" *A strong linear relationship was observed between the applied torque and measured strain, providing a method to measure gearbox usage.*"
* * *
**Reviewer#1, Comment # 22: Line 307: The ring gear does not rotate, so I'm not sure what is meant here.**

**Author response:** Sorry, we meant planetary gear's rotation and ring gear's strain response. The manuscript has been changed as:

"*DFOS effectively monitored the planetary gear's rotational direction, ring gear strain response, and potential temperature variations in real time.*"

**Reviewer#1, Comment # 23: Line 312: I see where the Abstract previously referred to "improved mechanical performance" is finally elaborated upon here in the form of "optimizing tooth design". I will admit though I'm having a hard time picturing how measuring ring gear strain could be used to optimize tooth design – unless multiple fibers were installed across the tooth facewidth and used to measure the tooth load distribution (i.e. Khbeta). With one fiber, I can see how tooth-to-tooth variations could be monitored. Can the authors elaborate upon what they mean here? Also, why is this aspect of the system only mentioned in the Abstract and the Conclusions, and not in the main body of the manuscript?**

**Author response:** We appreciate the reviewer's comment. We note that the improved mechanical performance is due to the measurement of strain on the ring gear when the internal gears rotate under different torque inputs. This measurement can be compared with the model, providing valuable feedback for its refinement.

Additionally, this experiment serves as a technology validation. DFOS can measure 50 meters—or even over 100 meters—of optical fiber with high spatial resolution, enabling the fiber to be wrapped around ring gears in multiple loops. Moreover, the optical fiber has been tested using a mesh design (https://ieeexplore.ieee.org/abstract/document/10076877) to measure the entire surface strain through interpolation. This unique feature allows DFOS to offer design feedback.

Since this application demonstrates potential rather than providing direct experimental results, we did not discuss it in the main body. Nonetheless, the DFOS results show that we can track tooth-to-tooth variations, which informed the design of the optical fiber installation at specific locations for enhanced design feedback.
* * *
**Reviewer#1, Comment # 24: Line 314: I also recommend revising or softening this sentence, especially the part about "significantly lowering operation and O&M costs".**

**Author response:** Thanks for the comment. The manuscript has been modified as:

*"Additionally, the straightforward installation process reduces the need for specialized training and accelerates deployment, demonstrating the potential to lower operating and O&M costs."*
* * *
**Reviewer#1, Comment # 25: Line 32: This sentence refers to "bearing fractures". I believe "bearing failures" is better in this context.**

**Author response:** Thanks for this comment. The manuscript has been changed as suggested.
* * *
**Reviewer#1, Comment # 26: Line 56: Please use correct inline citation style, in this case "Gutierrez Santiago et al. (2022)".**

**Author response:** Thanks for this comment. The manuscript has been changed as suggested.
* * *
**Reviewer#1, Comment # 27: Line 78: Please add reference for citation #27 and use consistent citation style.**

**Author response:** Thanks for catching this point. We have changed the citation and reference. The reference is:

"*Barter, G. E., Sethuraman, L., Bortolotti, P., Keller, J., and Torrey, D. A.: Beyond 15 MW: A cost of energy perspective on the next generation of drivetrain technologies for offshore wind turbines, Appl Energy, 344, 121272, https://doi.org/https://doi.org/10.1016/j.apenergy.2023.121272, 2023*."
* * *
**Review 2:**

**Health monitoring and fault detection are crucial research areas for enhancing the management of wind turbines. This manuscript presents a novel approach to monitoring wind turbine gearbox operation using high-resolution Distributed Fiber Optic Sensing (DFOS). The authors effectively demonstrate the method's capability in measuring strain profiles along a planetary gearbox, providing valuable insights into mechanical behavior and load distribution.**

**The application of DFOS for wind turbine gearbox monitoring is a promising technique that offers high spatial resolution and accuracy. The authors provide a thorough validation of their approach through load tests, demonstrating the accuracy and reliability of DFOS. The study highlights the potential of DFOS for early fault detection, improved mechanical performance, and more reliable wind turbine operations.**

**However, I would like to see a more detailed explanation of the temperature effects on the optical fiber. The current discussion is somewhat unclear, and it is not evident how the temperature effects were mitigated using a high-pass filter. A longer description and explanation of this aspect would strengthen the manuscript.**

**Overall, the manuscript is well-written, and the results are clearly presented. With minor revisions to address the temperature effects discussion, I recommend publication.**

We appreciate the comment from the reviewer. The temperature effect on the optical fiber was removed by a high-pass filter with the assumption that the thermal change is a slow process compared to the mechanical strain which affected by the rotation of the sun gears. The 0.01 Hz high pass filter was selected to remove the drift for better analysis of the rotational strain effect. It is not saying the thermal effect was 100% removed, but the assumption was that the temperature variation within 100 seconds is much smaller than the strain changes.

The description of the thermal effect after Figure 3 was modified as:

"*As the test progressed, a shift toward darker red regions indicated an increase in measured strain. However, since this strain was derived directly from the frequency shift, it still reflected a combination of both temperature and mechanical effects. The darker red horizontal strips were attributed to rising temperatures caused by frictional heat generated between the gears, while the diagonal strips were linked to the mechanical strain from the meshing of the rotational gears. Overall, the strain distribution and rotational behavior of the gears were clearly tracked throughout the test.*"

In the section of the "4.2 Thermal effect", the manuscript was changed to:

"*Temperature changes were also shown in the strain data, as shown in **Error! Reference source not found.**, which illustrates the response at a single location (bottom of the gearbox in **Error! Reference source not found.** or 4.62 m in Figure 1). Before 9:15 am, the strain exhibited a downward drift of approximately 30 microstrain. This is due to the gearbox having run overnight and it was stopped for this test and the drops indicating a cooling effect of about 3 °C. After 9:15 am, the measured strain, which includes mechanical strain and thermal, drifted by approximately 150 microstrain at 12:15, equivalent to a 15 °C temperature increase, caused by heat generated during operation. The maximum drift rate occurred between 9:19 and 9:41 am, coinciding with the transition from idle to maximum speed.*

*Because this test focuses on mechanical strains and the thermal effect due to friction was assumed to be a slow effect on the measured data, the slow drift temperature change shall be removed from the overall measured strain. The large drift can be removed by high-pass filter to isolate dynamic strain variations from slower thermal processes. In this research, a 0.01 Hz (100 seconds period) was applied to the measured strain. Although the temperature was not completely removed, due to the small fraction of thermal change generated in 100 seconds, the thermal induced drift became negligible, as shown in Figure 5 and Figure 6, highlights the strain behavior under varying loads."*

**Reviewer 3**

**This manuscript provides an interesting application of distributed fiber optic sensors for monitoring of gearbox strains.**

**Overall the paper is well written, structured and presented. I suggest revising figures**

**Figures 3, 5, 6, 8 and check legends and units, font size and printing quality.**

We appreciate this comment from the reviewer. We modified the mentioned figures, and they can be seen in the supplement document below:

[Figure]

Figure 1 Waterfall representation of the strain profile over time for the circumference of the ring rear. The vertical axis denotes distance in meters, while the color gradient indicates strain in micro-strain. (a) Strain profile throughout the entire test period. (b) A magnified view of the strain profile corresponding to the region outlined by the white dashed square in the left panel.

[Figure]

Figure 2 Waterfall representation of the strain profile over time for the circumference of the ring rear. The vertical axis denotes distance in meters, while the color gradient indicates strain in micro-strain. (a) Strain profile throughout the entire test period. (b) A magnified view of the strain profile corresponding to the region outlined by the white dashed square in the left panel.

[Figure]

Figure 3 The waterfall of the strain response during the load test after 0.01 Hz high pass filter in period of 30 seconds when the load is at 100% torque, time domain shows strain profile collected with 12.5 Hz.

[Figure]

Figure 4 The strain response after 0.01 Hz high pass at location 4.62m (one tooth's strain response). The dash line shows the timing of each loading event listed in table 2. The color background shows the percentage of the rated torque input from 17% to 100%.

[Figure]

Figure 5 (a) An example of collected strain at 5 location with time and (b) the peak-peak strain at 49 locations on the ring gear for 6 torque load inputs.

[Figure]

Figure 6 (a) An example of collected strain at 5 location with time and (b) the peak-peak strain at 49 locations on the ring gear for 6 torque load inputs.

---

## Referee Report (RR1)

In this paper, the authors describe the interesting use of distributed fiber optic sensors for measurement of gearbox torque and temperature measurement. Such a system can be used for gearbox usage monitoring and has potential for health monitoring. I appreciate the authors' recent revisions, but I still offer the following comments for improvement and greater clarity.

Abstract

- Lines 13-17: I did not notice it during the first review, but the phrase "…captured millimeter-scale distributed strain profiles…" may be a little confusing, as this I think implies the strain being measured is on the order of millimeters, rather than being measured every few millimeters. That is the quantity being measured is related to strain (distance) and is measured at a fine spatial resolution (distance), so care is needed here. This is stated later in the paragraph with "…measured strain across all monitored locations …, with data collected every 2.6 mm". After rereading (including similar text in the Conclusions), I suggest the following wording for these 2 sentences "…we measured circumferential strain from planetary gear passage every 2.6 mm around the ring gear under different input torque levels. Our results show accurate identification of planet gear locations in real time and rotation speed (10.42 revolutions per minute) with a strong linear correlation ($R2$=0.9997) between applied torque and measured strain across all 2,500 measured locations."
- Lines 21-22: I understand the authors' perspective and recent changes here, but the authors' response is better stated than the current sentence. That is, the DFOS alone doesn't "enhance structural integrity and operational efficiency" – or really what "operational efficiency" means in this case, which I tend to interpret literally as power transmission efficiency. I recommend instead a sentence similar to that described in the authors' response, such as "The approach offers a scalable and practical solution for early fault detection and support of design validation, and when combined with modeling can lead to a more durable and optimized design."

Introduction

- Line 32: I appreciate the authors' recent change here. However, rather than quoting the number of replacements, which in the case of 2,000 per year is specific to the US market, it makes more sense to simply say "Approximately 1% to 2% of gearboxes are replaced annually (Haus et al., 2024)."
- Line 63: The phrase "…measure strain continuously at millimeter-scale resolution around…" may seem contradictory or confusing as described above. I suggest simply "…measure strain every few millimeters around…".

Gearbox and Distributed fiber optic sensor

- Line 99: The phrase "…enabling continuous sensing at millimeter-scale resolution" may seem contradictory or confusing as described above. I suggest simply "…enabling sensing every few millimeters along the length of the optical fiber".

Results and Discussion

- Line 169: I appreciate the authors' revisions; however, I might recommend a slight modification to "…caused by frictional heat generated by shearing of the oil between the gear teeth and in the bearings, while…" I ask the authors to please though check my interpretation of their

statement for accuracy – I believe it's really a matter of stating the main cause(s) of temperature rise in a gearbox.

- Line 238: I appreciate the authors' response regarding "ratcheting". Considering that, I recommend adding the provided short description to the term such as "…ratcheting (i.e. the progressive, incremental deformation that occurs in a material when it is subjected to cyclic loading), inadequate lubrication…".

- Line 291: Similar to previous comments about the meaning of "operational efficiency", here I believe this refers to making O&M easier in general and reducing O&M costs. So, I recommend this be simplified to "…enabling proactive maintenance. This is particularly important for offshore wind farms, where reducing downtime and O&M costs is critical."

Conclusion

- Lines 309-311: Similar to previous sentences, I think the phrase "…millimeter-scale spatial resolution and a strain measurement accuracy of approximately 1 microstrain at a sampling rate of 12.5 Hz, with approximately 2,500 data points collected simultaneously around the gearbox" can be more clearly stated as "…a strain measurement accuracy of approximately 1 microstrain at a sampling rate of 12.5 Hz simultaneously over approximately 2,500 measurement locations spaced every 2.6 mm around the gearbox circumference".

- Line 325: Related to the changes in the Abstract, I think a more accurate phrase than "…optimizing tooth design for improved mechanical performance and refining gearbox control systems" is "…optimizing tooth design for improved reliability and refining turbine control systems".

Minor grammatical comments

- Line 292: Should be "the current DFOS system also has limitations."

---

## Author Response (AR2)

**Original Manuscript ID:** wes-2025-1

**Original Article Title:** "Wind Turbine Gearbox Operation Monitoring Using High-Resolution Distributed Fiber Optic Sensing"

**To:** Wind Energy Science

**Re:** Response to reviewers

Dear Editor,

Thank you for allowing the resubmission of our manuscript, with an opportunity to address the reviewers' comments.

We appreciate the detailed and constructive comments and questions from the reviewers. We hope the revised manuscript addresses all the comments.

Best regards,

Linqing Luo on behalf of the authors

**Reviewer#1, Comment # 1: Lines 13-17: I did not notice it during the first review, but the phrase "…captured millimeter-scale distributed strain profiles…" may be a little confusing, as this I think implies the strain being measured is on the order of millimeters, rather than being measured every few millimeters. That is the quantity being measured is related to strain (distance) and is measured at a fine spatial resolution (distance), so care is needed here. This is stated later in the paragraph with "…measured strain across all monitored locations …, with data collected every 2.6 mm". After rereading (including similar text in the Conclusions), I suggest the following wording for these 2 sentences "…we measured circumferential strain from planetary gear passage every 2.6 mm around the ring gear under different input torque levels. Our results show accurate identification of planet gear locations in real time and rotation speed (10.42 revolutions per minute) with a strong linear correlation ($R^2$=0.9997) between applied torque and measured strain across all 2,500 measured locations."**

**Author response:** The authors appreciate the reviewer's comment and modify the manuscript accordingly.

We changed the text in the abstract from:

*"we captured millimeter-scale distributed strain profiles of the ring gear from planetary gears passage under different input torque levels. Our results show accurate identification of planet gear locations in real time and rotation speed (10.42 revolutions per minute) and a strong linear correlation between applied torque and measured strain across all monitored locations (R^2=0.9997), with data collected every 2.6 mm."*

To

*"we measured circumferential strain from planetary gear passage every 2.6 mm around the ring gear under different input torque levels. Our results show accurate identification of planet gear locations in real time and rotation speed (10.42 revolutions per minute), with a strong linear correlation (R² = 0.9997) between applied torque and measured strain across all 2,500 measured locations."*

*We also modified the conclusion from:*

*"This study introduces a novel approach to measuring fully distributed strain profiles along a planetary gearbox using DFOS. By measuring circumferential strain every 2.6 mm around the ring gear under varying input torque, DFOS achieves high-resolution strain mapping with an accuracy of approximately 1 microstrain at a 12.5 Hz sampling rate, collecting around 2,500 data points simultaneously around the gearbox."*

To

*"This study introduces a novel approach to measuring fully distributed strain profiles along a planetary gearbox using DFOS. By measuring circumferential strain every 2.6 mm around the ring gear under varying input torque, DFOS achieves high-resolution strain mapping with an accuracy of approximately 1 microstrain at a 12.5 Hz sampling rate, collecting around 2,500 data points simultaneously around the gearbox."*
* * *
**Reviewer#1, Comment# 2: Lines 21-22: I understand the authors' perspective and recent changes here, but the authors' response is better stated than the current sentence. That is, the DFOS alone doesn't "enhance structural integrity and operational efficiency" – or really what "operational efficiency" means in this case, which I tend to interpret literally as power transmission efficiency. I recommend instead a sentence similar to that described in the authors' response, such as "The approach offers a scalable and practical solution for early fault detection and support of design validation, and when combined with modeling can lead to a more durable and optimized design."**

**Author response:** The authors appreciate the reviewer's comment and modify the manuscript accordingly.

This sentence has been modified from:

*"The approach offers a scalable and practical solution for early fault detection, enhancing the ring gear's structural integrity and operational efficiency, addressing critical challenges in the wind energy sector."*

To

*"The approach offers a scalable and practical solution for early fault detection and support of design validation, and when combined with modeling can lead to a more durable and optimized design."*
* * *
**Reviewer#1, Comment # 3: Line 32: I appreciate the authors' recent change here. However, rather than quoting the number of replacements, which in the case of 2,000 per year is specific to the US market, it makes more sense to simply say "Approximately 1% to 2% of gearboxes are replaced annually (Haus et al., 2024)."**

**Author response:** The authors appreciate the review's comment and modify the manuscript accordingly.

This sentence has been modified from:

*"As of 2024, approximately 2,000 gearbox failures were reported annually."*

To

*"Approximately 1% to 2% of gearboxes are replaced annually."*
* * *
**Reviewer#1, Comment # 4: Line 63: The phrase "…measure strain continuously at millimeter-scale resolution around…" may seem contradictory or confusing as described above. I suggest simply "…measure strain every few millimeters around…".**

**Author response:** The authors appreciate the review's comment and modify the manuscript accordingly.

This sentence has been modified from:

*"To overcome these limitations, this study proposes using Distributed Fiber Optic Sensing (DFOS) to measure strain continuously at millimeter-scale resolution around the entire gearbox circumference."*

To

*"To overcome these limitations, this study proposes using Distributed Fiber Optic Sensing (DFOS) to measure strain every few millimeters around the entire gearbox circumference."*
* * *
**Reviewer#1, Comment # 5: Line 99: The phrase "…enabling continuous sensing at millimeter-scale resolution" may seem contradictory or confusing as described above. I suggest simply "…enabling sensing every few millimeters along the length of the optical fiber".**

**Author response:** The authors appreciate the review's comment and modify the manuscript accordingly.

This sentence has been modified from:

*"DFOS systems measure changes in light properties along an optical fiber, enabling continuous sensing at millimeter-scale resolution."*

To

*"DFOS systems measure changes in light properties along an optical fiber, enabling sensing every few millimeters along the length of the optical fiber."*
* * *
**Reviewer#1, Comment # 6: Line 169: I appreciate the authors' revisions; however, I might recommend a slight modification to "…caused by frictional heat generated by shearing of the oil between the gear teeth and in the bearings, while…" I ask the authors to please though check my interpretation of their statement for accuracy – I believe it's really a matter of stating the main cause(s) of temperature rise in a gearbox.**

**Author response:** The authors appreciate the comment. The suggested modification has been added to the manuscript.

This sentence has been modified from:

*"The darker red horizontal strips were attributed to rising temperatures caused by frictional heat generated between the gears, while the diagonal strips were linked to the mechanical strain from the meshing of the rotational gears."*

To

*"The darker red horizontal strips were attributed to rising temperatures caused by frictional heat generated by shearing the gear teeth when gears rotate, while the diagonal strips were linked to the mechanical strain from the meshing of the rotational gears."*

**Reviewer#1, Comment # 7: Line 238: I appreciate the authors' response regarding "ratcheting". Considering that, I recommend adding the provided short description to the term such as "…ratcheting (i.e. the progressive, incremental deformation that occurs in a material when it is subjected to cyclic loading), inadequate lubrication…".**

**Author response:** The authors appreciate the comment. The suggested modification has been added to the manuscript.

*"However, as the gearbox ages, deviations from this linearity may appear due to factors such as material fatigue, ratcheting (i.e. the progressive, incremental deformation that occurs in a material when it is subjected to cyclic loading), inadequate lubrication, or tooth wear, such as micropitting."*
* * *
**Reviewer#1, Comment # 8: Line 291: Similar to previous comments about the meaning of "operational efficiency", here I believe this refers to making O&M easier in general and reducing O&M costs. So, I recommend this be simplified to "…enabling proactive maintenance. This is particularly important for offshore wind farms, where reducing downtime and O&M costs is critical."**

**Author response:** The authors appreciate the comment. The suggested modification has been added to the manuscript.

This sentence has been modified from:

*"These features support comprehensive monitoring of the gearbox, enabling proactive maintenance and improving operational efficiency. This is particularly important for offshore wind farms, where reducing downtime and O&M costs is critical.*

To

*"These features support comprehensive monitoring of the gearbox, enabling proactive maintenance. This is particularly important for offshore wind farms, where reducing downtime and O&M costs is critical.*
* * *
**Reviewer#1, Comment #9: Lines 309-311: Similar to previous sentences, I think the phrase "…millimeter-scale spatial resolution and a strain measurement accuracy of approximately 1 microstrain at a sampling rate of 12.5 Hz, with approximately 2,500 data points collected simultaneously around the gearbox" can be more clearly stated as "…a strain measurement accuracy of approximately 1 microstrain at a sampling rate of 12.5 Hz simultaneously over approximately 2,500 measurement locations spaced every 2.6 mm around the gearbox circumference".**

**Author response:** We appreciate the reviewer for the suggestion.

The manuscript has been modified from:

*"This study introduces a novel approach to measuring fully distributed strain profiles along a planetary gearbox using DFOS. Leveraging OFDR technology, DFOS achieves millimeter-scale spatial resolution and a strain measurement accuracy of approximately 1 microstrain at a sampling rate of 12.5 Hz, with approximately 2,500 data points collected simultaneously around the gearbox."*

*To*

*"By measuring circumferential strain every 2.6 mm around the ring gear under varying input torque, DFOS achieves high-resolution strain mapping with an accuracy of approximately 1 microstrain at a 12.5 Hz sampling rate, collecting around 2,500 data points simultaneously around the gearbox."*
* * *
**Reviewer#1, Comment # 10: Line 325: Related to the changes in the Abstract, I think a more accurate phrase than "…optimizing tooth design for improved mechanical performance and refining gearbox control systems" is "…optimizing tooth design for improved reliability and refining turbine control systems".**

**Author response:** We appreciate the reviewer for the suggestion.

The manuscript has been modified from:

*"This capability can contribute to optimizing tooth design for improved mechanical performance and refining gearbox control systems."*

*To*

*"This capability can contribute to optimizing tooth design for improved reliability and refining turbine control systems."*
* * *
**Reviewer#1, Comment # 11: Line 292: Should be "the current DFOS system also has limitations."**

**Author response:** We appreciate the reviewer for the suggestion. The "have" is modified to "has".